# Isolation of Bioactive Compounds and Antioxidant Activity Evaluation of *Crataegus monogyna* Leaves via Pulsed Electric Field-Assisted Extraction

**DOI:** 10.3390/plants14152262

**Published:** 2025-07-22

**Authors:** Vasiliki Papazidou, Ioannis Makrygiannis, Martha Mantiniotou, Vassilis Athanasiadis, Eleni Bozinou, Stavros I. Lalas

**Affiliations:** 1Department of Chemical Engineering, University of Western Macedonia, 50100 Kozani, Greece; vpapazidou@gmail.com; 2Department of Food Science and Nutrition, University of Thessaly, 43100 Karditsa, Greece; ioanmakr1@uth.gr (I.M.); mmantiniotou@uth.gr (M.M.); vaathanasiadis@uth.gr (V.A.); empozinou@uth.gr (E.B.)

**Keywords:** hawthorn, antioxidants, polyphenols, HPLC-DAD, response surface methodology, extraction optimization, phenolic characterization, sustainable extraction methods

## Abstract

*Crataegus monogyna*, commonly known as hawthorn, is a valuable plant in pharmaceutical production. Its flowers, leaves, and fruits are rich in antioxidants. This study explores the application of pulsed electric field (PEF) for enhanced extraction of bioactive compounds from *C. monogyna* leaves. The liquid-to-solid ratio, solvent composition (ethanol, water, and 50% *v/v* aqueous ethanol), and key PEF parameters—including pulse duration, pulse period, electric field intensity, and treatment duration—were investigated during the optimization process. To determine the optimal extraction conditions and their impact on antioxidant activity, response surface methodology (RSM) with a six-factor design was employed. The total polyphenol content in the optimized extract was 244 mg gallic acid equivalents/g dry weight, while individual polyphenols were analyzed using high-performance liquid chromatography coupled with a diode array detector (HPLC-DAD). Furthermore, antioxidant activity was assessed using ferric-reducing antioxidant power (FRAP) and DPPH radical scavenging assays, yielding values of 3235 and 1850 μmol ascorbic acid equivalents/g dry weight, respectively. Additionally, correlation analyses were conducted to evaluate the interactions between bioactive compounds and antioxidant capacity. Compared to other extraction techniques, PEF stands out as an eco-friendly, non-thermal standalone method, offering a sustainable approach for the rapid production of health-promoting extracts from *C. monogyna* leaves.

## 1. Introduction

Polyphenols constitute a class of plant secondary metabolites characterized by great structural and functional diversity [1]. They are found in various plant-derived sources, such as fruits (berries, watermelon, apples, grapes, etc.), vegetables (soybeans, onions, etc.), and cereals, as well as in beverages like coffee, red wine, and juices [1,2]. Polyphenolic compounds encompass phenols, flavonoids, phenolic acids, proanthocyanidins, tannins, lignans, coumarins, and stilbenes, which occur in different parts of plants, such as leaves, flowers, roots, and shoots [3,4]. They have attracted considerable scientific interest for their potent antioxidant activity, primarily because they can reduce oxidative stress and neutralize free radicals [5], effects which, in turn, contribute to the prevention of chronic diseases such as cardiovascular disorders, neurodegenerative conditions, and certain types of cancer [1,6]. The food, pharmaceutical, and cosmetic industries have begun to utilize phenolic compounds more frequently in recent years because of their antibacterial, anti-inflammatory, and antioxidant characteristics [3,7]. The development and improvement of technologies for extracting and purifying these compounds from plant materials, foods, and food-industry by-products reflect the increasing interest in these compounds and their potential uses [3].

Among plants rich in polyphenols is *C. monogyna* (CM) Jacq., commonly known as the hawthorn [8,9]. Its scientific name derives from the Greek word “kràtaigos,” meaning “strength and robustness,” in reference to its hard, durable wood [10]. Hawthorn is a deciduous shrub with distinctive white flowers and red berries [8]. It is endemic to temperate regions of the Northern Hemisphere, including Europe, Asia, and North Africa, and has a long history of use in folk medicine for its cardioprotective and neuroprotective properties [8,9]. The fruits, leaves, and flowers of CM are rich in phenolic compounds, including flavonoids such as rutin, quercetin 3-*D*-galactoside (hyperoside), and vitexin, as well as other constituents such as vitamin C, saponins, tannins, cardiotonic amines (e.g., phenylethylamine, tyramine), procyanidins, triterpenoid acids (e.g., ursolic acid), and purine derivatives (e.g., adenosine, guanine) [4,8,11,12,13]. The bioactive profile of hawthorn has been linked to antioxidant, hypolipidemic, anti-inflammatory, and neuroprotective activities, underscoring its potential for applications in food, pharmaceutical, and cosmetic formulations [8,9,14].

Extraction is a pivotal step in the chemical analysis of plant samples, necessary for sample preparation and the isolation of bioactive compounds from plant tissue [15]. Various extraction techniques for bioactive compounds are reported in the literature, with solid–liquid extraction among the most common for isolating plant antioxidants [16]. Today, extraction methods are broadly classified as conventional or non-conventional [17]. In conventional approaches, bioactive substances are removed from plant material using traditional solvents (with optional heating) [16]. The sample is first homogenized and then immersed in a single solvent or solvent mixture under continuous agitation, allowing the target compounds to diffuse and transfer into the solvent [16,18]. Traditional methods—including percolation, maceration, and Soxhlet extraction—rely on straightforward procedures to isolate specific constituents and produce crude extracts [15]. These extracts may be used directly or formulated into herbal medicines, dietary supplements, and cosmetic ingredients [15,19,20,21]. However, conventional techniques suffer from drawbacks such as long extraction times, low yield, loss of nutrients, high solvent and energy consumption, and often require multiple extraction steps [22,23]. Moreover, heating can degrade or alter heat-sensitive phytochemicals [22].

By contrast, modern extraction methods achieve higher yields and greater selectivity [15]. These methods encompass ultrasound-assisted extraction, microwave-assisted extraction, pressurized liquid extraction, supercritical fluid extraction, pulsed electric field extraction, and enzymatic extraction [15,23]. The use of “green” solvents further transforms these approaches into environmentally friendly extraction techniques [24]. In recent years, numerous laboratory-scale studies employing such green methods have yielded high-value extracts from plant materials [25].

To overcome the limitations of both conventional and current green methods, advanced non-thermal technologies have been developed, among which PEF extraction stands out as a promising and efficient alternative. PEF applies short electric pulses that permeabilize cell membranes by creating pores [26]. Its advantages over other techniques include ultrashort processing times (nanoseconds to milliseconds), enhanced extraction efficiency (greater membrane permeability), reduced energy consumption, preservation of cellular structure, and higher quality of the final extracts [26].

Optimization of extraction parameters is essential to maximize yield while preserving the integrity of bioactive compounds. Response surface methodology (RSM)—a combined statistical and mathematical approach—provides an effective framework for modeling and optimizing extraction processes [27]. RSM enables simultaneous evaluation of multiple independent variables to identify optimal conditions with fewer experimental runs than traditional methods.

The objective of this study is to identify optimal conditions for PEF-assisted polyphenol extraction from CM leaves utilizing RSM. Key extraction parameters, such as electric field intensity, pulse duration, solvent concentration, and extraction time, will be tuned by RSM. The Folin–Ciocalteu assay will quantify total polyphenol content (TPC), high-performance liquid chromatography coupled with a diode array detector (HPLC-DAD) will identify individual phenolic compounds, and DPPH radical scavenging activity and ferric-reducing antioxidant power (FRAP) assays will evaluate antioxidant activity.

## 2. Results and Discussion

### 2.1. Extraction Parameters Optimization

Regarding sustainability, extraction methods such as PEF treatment require lower energy use compared to conventional processes [28]. The distinctive brevity of the PEF-based extraction method aids in energy conservation. Furthermore, many processing parameters in conjunction with the extraction procedure may considerably influence both the extract yield and antioxidant efficacy [29]. The essential PEF parameters include electric field strength, pulse width, pulse duration, and treatment duration. Furthermore, all extractions are influenced by the solvent composition, owing to the varying polarity of bioactive compounds and the liquid-to-solid ratio. Consequently, all these parameters merit examination. Table 1 illustrates the impact of the investigated variables on the analyzed responses, whereas Table 2 displays the ANOVA results applied to the RSM quadratic polynomial model.

### 2.2. Model Analysis

Regression models relevant to the extraction process are shown in Equations (1)–(3), which predict the following critical response variables: TPC, FRAP, and DPPH radical scavenging activity. The complex interplay between the various experimental variables is highlighted by the fact that each equation contains both linear and quadratic components, as well as an interaction component. Only important terms are included in the models. Extraction efficiency is emphasized by the regression models as being affected by solvent composition, extraction temperature, and time. Optimal conditions for optimum antioxidant output are suggested by the linear and quadratic components, which show nonlinear correlations among elements. An important element that affects the FRAP and DPPH equations is the extraction duration (*X*_4_). There needs to be careful parameter optimization because the presence of interaction terms shows that antioxidant potential is affected by the cumulative impact of multiple variables. Bioactive compounds, such as antioxidants, can be dissolved into the solvent in greater amounts when the extraction time is prolonged. An ideal range for extraction time is indicated by the presence of quadratic terms (*X*_1_^2^, *X*_4_^2^, *X*_5_^2^, and *X*_6_^2^) and interactions (*X*_1_*X*_4_, *X*_1_*X*_6_ and *X*_1_*X*_3_, *X*_4_*X*_6_, etc.); a shorter duration may limit compound release, while a longer duration may cause degradation or reduced efficiency.
*TPC* = 337.87 − 654.38*X*_1_ − 0.02*X*_2_ + 5.28*X*_3_ + 2.21*X*_4_ − 4.06*X*_5_ − 13.14*X*_6_ + 371.55*X*_1_^2^ − 0.04*X*_3_^2^ − 0.01*X*_4_^2^ + 0.04*X*_5_^2^ + 0.19*X*_6_^2^ − 2.74*X*_1_*X*_4_ + 2.73*X*_1_*X*_5_ + 5.68*X*_1_*X*_6_ − 0.0003*X*_2_*X*_4_ + 0.004*X*_2_*X*_6_ − 0.053*X*_3_*X*_6_ + 0.007*X*_4_*X*_5_ + 0.034*X*_4_*X*_6_(1)
*FRAP* = 6095.75 − 7218.12*X*_1_ − 0.96*X*_2_ − 18.06*X*_3_ − 3.17*X*_4_ − 17.22*X*_5_ + 3.43*X*_6_ + 3485.51*X*_1_^2^ − 0.21*X*_4_^2^ + 0.10*X*_5_^2^ + 22.20*X*_1_*X*_3_ + 12.17*X*_1_*X*_5_ + 0.008*X*_2_*X*_3_ − 0.0037*X*_2_*X*_4_ + 0.019*X*_2_*X*_5_ + 0.016*X*_2_*X*_6_ + 0.039*X*_3_*X*_4_ − 0.057*X*_3_*X*_5_ − 0.387*X*_3_*X*_6_ + 0.057*X*_4_*X*_5_ + 0.434*X*_4_*X*_6_ − 0.339*X*_5_*X*_6_(2)
*DPPH* = 6223.05 − 9594.53*X*_1_ − 0.86*X*_2_ − 21.94*X*_3_ − 4.95*X*_4_ − 13.75*X*_5_ + 5.77*X*_6_ + 5302.51*X*_1_^2^ − 0.15*X*_4_^2^ + 0.08*X*_5_^2^ + 0.46*X*_1_*X*_2_ + 16.22*X*_1_*X*_3_ + 0.007*X*_2_*X*_3_ − 0.0015*X*_2_*X*_4_ + 0.011*X*_2_*X*_5_ + 0.068*X*_3_*X*_4_ + 0.054*X*_3_*X*_5_ − 0.10*X*_3_*X*_6_ + 0.116*X*_4_*X*_5_ + 0.250*X*_4_*X*_6_ − 0.352*X*_5_*X*_6_(3)


The 3D plots in Figure 1, Figure 2 and Figure 3 elucidate the combined influence of extraction parameters on three key antioxidant metrics: TPC, FRAP, and DPPH. Overall, these visualizations highlight the nonlinear behavior of bioactive compound recovery and underscore the importance of parameter optimization in PEF-assisted extraction.

Figure 1 demonstrates that TPC is most responsive to moderate electric field strength and mid-range ethanol concentrations. Excessive ethanol (100%) notably suppressed TPC yield, reinforcing that solvent polarity must match the compound profile to ensure efficient extraction. Additionally, high liquid-to-solid ratios and shorter extraction times under optimal PEF conditions significantly enhanced polyphenol release, likely due to improved cell disruption and diffusion kinetics.

Figure 2 extends these observations to FRAP values, revealing that longer pulse periods combined with moderate solvent concentrations amplify antioxidant power. Interactions between pulse parameters (duration and period) and physical conditions (extraction time and solvent composition) play a pivotal role in maximizing FRAP. The data suggest that electron-donating phenolics may be more susceptible to variations in time and electrical input, supporting the need for precise control during process scale-up.

Figure 3 offers a more complex picture of DPPH radical scavenging activity, where the response was less predictable across variable combinations. While longer pulse periods and intermediate ethanol levels often led to enhanced radical scavenging performance, the model also identified parameter regions where lower or higher values had better outcomes. This complexity may stem from the diverse structural properties of antioxidant compounds and their varying sensitivity to treatment conditions.

Table 3 consolidates the predictive modeling results, presenting the optimal conditions for each response with corresponding desirability values—0.9978 for TPC, 0.9930 for FRAP, and 0.9980 for DPPH—indicating robust statistical fit. However, a key observation is the divergence in parameter settings required to optimize each assay: for instance, FRAP peaked under lower ethanol concentration and shorter extraction time, whereas TPC favored higher field strength and prolonged pulse duration. This divergence highlights the trade-offs inherent in multi-response optimization and justifies the use of statistical tools like RSM and stepwise regression to navigate the multidimensional design space.

### 2.3. Influence of Extraction Parameters on Assays via Pareto Plot Analysis

Orthogonal estimates serve as a prevalent statistical instrument for assessing the relative significance of many components in a Pareto plot, aiming to diminish intercorrelation. This strategy facilitates the identification of which factors exerted the most significant influence on a specific outcome by maintaining the separation of estimates. Regression analysis and experimental design often utilize orthogonal estimates to improve the accuracy of parameter estimation. They are effective in reducing the probability of bias and ensuring that inter-variable interactions do not influence the calculated effects. A normalized Pareto plot was utilized to evaluate the principal effects and their interactions based on statistical significance (*p* < 0.05). Figure 4 illustrates the independent variables and their interactions that influenced the results under investigation. According to Pareto plot analysis, the liquid–solid ratio (*X*_5_) had a positive effect on TPC, while most other factors had a minimal effect on extraction efficiency. The concentration of the solvent adversely impacted both FRAP and DPPH, reaffirming the significance of the polarity of the target compounds and the composition of the solvent in their recovery efficiency. Regarding PEF parameters, electric field strength (*X*_1_) had no significant effect on any response, pulse period (*X*_2_) significantly affected only FRAP and DPPH, and pulse duration (*X*_3_) negatively affected only DPPH.

### 2.4. Principal Component Analysis (PCA) and Multivariate Component Analysis (MCA)

The associations between the tests and extraction conditions were further analyzed through correlation analyses, including PCA and MCA, as depicted in Figure 5 and detailed in Table 4, respectively. The objective of PCA, a dimensionality reduction technique, is to condense a dataset with numerous interrelated factors into a smaller collection of uncorrelated variables. These pieces are meticulously selected to represent the data’s most significant variations. Numerous fields utilize PCA for data preprocessing, visualization, and investigation. The MCA further clarifies the interdependence among variables. The primary advantage of this method is determining the strength of the positive or negative correlation between the examined variables. Correlation studies were performed to determine the correlations between the variables and TPC, FRAP, and DPPH within the framework of PCA. The data indicated that PC1 and PC2 contributed 78.1% and 19.9%, respectively, accounting for 98% of the total variance. The analysis was found to be substantially affected by the independent variables. According to PCA, *X*_3_, *X*_4_, and *X*_6_ exhibited a negative correlation with all the other responses, while *X*_1_, *X*_2_, and *X*_5_ showed a positive correlation with TPC but a negative one with both antioxidant assays. Previous studies also demonstrated that an increase in ethanol concentration restricted high recoveries on TPC, FRAP, and DPPH [30]. The results of MCA also reinforced the claim that TPC opposes FRAP, and DPPH. More specifically, TPC demonstrated a moderate correlation (~0.60) with FRAP but a particularly low correlation with DPPH (~0.46), while FRAP and DPPH had a strong correlation with each other (~0.92).

### 2.5. Partial Least Squares (PLS) Analysis

A PLS model was utilized to ascertain the importance of the extraction parameters and identify the optimal ones. PLS analysis (Figure 6A) was employed to generate a correlation loading map that clearly illustrates the extraction conditions of CM leaves. Meanwhile, plot (B) illustrates the Variable Importance Plot (VIP), which underscores the significance of each predictor variable in the PEF extraction approach. The red dashed line serves as a reference for the 0.8 significance threshold, emphasizing the relative impact of each variable within the model. The VIP plot shows that the parameters most affecting performance are pulse period, solvent composition, and extraction time. Solvent composition had the highest score, followed by the squared term (*X*_4_^2^). In PLS, the influences of the parameters on the extraction efficiency are clearly shown, thereby enabling the formulation of optimal conditions. These were formulated as follows: 1 kV/cm electric field strength, a pulse period of 1000 μs, a pulse duration of 75 μs, and an extraction time of 10 min, while the optimal extraction solvent was 19% *v/v* aqueous ethanol and the liquid-to-solid ratio was 70 mL/g.

The robust association between the experimental findings and PLS model predictions is evidenced by a high correlation value of 0.997 and a significant coefficient of determination (R^2^) of 0.994. Furthermore, the minimal *p*-value (<0.0001) confirms that the differences between observed and predicted values are statistically insignificant.

### 2.6. Comparative Analysis of Extraction Techniques

To validate the efficiency of PEF extraction under optimal conditions, three additional extraction protocols were performed: a control extraction (No-PEF), a conventional stirring extraction (STE), and a green ultrasound-assisted extraction (UAE). All methods applied identical liquid–solid ratios, solvent concentration, and extraction time to ensure consistency.

The No-PEF control involved passive solvent exposure, where the CM leaf powder was immersed in solvent for 10 min without electric stimulation, serving as a baseline for solvent–solid interaction. STE offered a conventional mechanical alternative, while the UAE method introduced a green approach using ultrasound waves at 37 kHz in pulse mode—selected to balance effective cavitation and sample integrity.

Figure 7 illustrates the extraction yields of the four extraction techniques, and Table 5 details the individual compounds identified in the corresponding extracts using high-performance liquid chromatography coupled with a diode array detector (HPLC-DAD). While PEF extraction demonstrated strong performance and clearly outperformed the No-PEF control—highlighting the role of electric stimulation in enhancing compound release—STE yielded the highest overall recovery of both polyphenols and antioxidant activity. UAE ranked third, whereas the No-PEF passive extraction consistently produced the lowest values, reinforcing that mere solvent interaction is insufficient for optimal recovery. These results confirm that mechanical agitation in STE and electrical disruption in PEF both significantly improve polyphenol accessibility from CM leaves.

The optimal TPC obtained by PEF was 244 mg GAE/g dw, which represents a notably high yield for CM leaves, exceeding those reported for CM fruits by a factor of 5.5 in a previous study [8], while Bahorun et al. [31] determined a TPC 5.1 times lower for CM fruit than that observed in the current study. Furthermore, antioxidant activity as measured via FRAP and DPPH assays reached 3235 and 1850 μmol AAE/g dw, respectively, indicating a strong radical scavenging potential in the leaf extracts compared to previously reported values.

Figure 8 illustrates a representative chromatograph of all four extracts, while Table 6 provides the equations used for quantification, along with R-squared (R^2^) values, retention time (RT), UV_max_ of each compound, limit of detection (LOD), and limit of quantification (LOQ). Separation and retention times confirm the identification of compounds via reference standards. Pelargonin chloride, cyanidin 3-glucoside chloride, and quercetin 3-*D*-galactoside (hyperoside) were conclusively identified through comparison with authenticated standards, and their corresponding UV-Vis spectra are presented in Figure A1 as supplementary validation. The spectral profiles exhibit strong alignment with the expected absorbance maxima: pelargonin chloride showed peaks ranging from 265 to 645 nm, cyanidin 3-glucoside displayed its characteristic λ_max_ at 516 nm, and quercetin 3-*D*-galactoside registered a prominent peak at 352 nm. Minor spectral shifts between standards and extracts—attributable to matrix interactions—did not affect compound identification, as core spectral features remained intact. These results confirm the chemical integrity and structural identity of key polyphenols within CM leaf extracts and reinforce the accuracy and reliability of HPLC-DAD quantification at their respective UV_max_ wavelengths.

The main compound found in the extracts of CM leaves is pelargonin chloride. This compound is very prevalent in berry fruits and possesses antioxidant and anti-inflammatory activities [32]. The amount found in CM leaves is remarkable, considering that only up to 0.01 mg/g pelargonin chloride has been found in strawberry fruits [33]. Cyanidin 3-glucoside, which is typically used as a food colorant, also displays health benefits, such as antioxidant, anticancer, and anti-inflammatory effects [34]. Cyanidin 3-glucoside is a typical anthocyanin found in berry fruits like mulberries and strawberries [33,35]. Zannou et al. [36] determined approximately 1.2 mg/g of this particular anthocyanin is found in blackberry fruit, an amount significantly lower than the one determined in the current study. The third compound identified through HPLC-DAD was quercetin 3-*D*-galactoside, a flavonoid with strong antioxidant activity that is also common in berry fruits [37]. Quercetin 3-*D*-galactoside was also found in CM leaves by Kirakosyan et al. [38], in a quantity of 0.387 mg/g dw. Researchers from another study [39] identified this flavonoid in *Moringa oleifera* leaves at a quantity almost three times lower than that in CM leaves. These findings indicate that CM leaves are carriers of potent antioxidant compounds of pharmacological interest that could be isolated and utilized in various applications.

## 3. Materials and Methods

### 3.1. Chemicals and Reagents

Ethanol (99.8%), Folin–Ciocalteu’s reagent, and gallic acid (97%) were obtained from Panreac Co. (Barcelona, Spain). Acetonitrile (99.9%) was purchased from Labkem (Barcelona, Spain). Hydrochloric acid (37%), 2,2-diphenyl-1-picrylhydrazyl (DPPH), 2,4,6-tris(2-pyridyl)-s-triazine (TPTZ) (≥98%), and all polyphenolic standards for the HPLC determination (at least 97% purity or higher) were obtained from Sigma-Aldrich (Darmstadt, Germany). Formic acid (99.8%), sodium carbonate (anhydrous, 99.5%), and rutin (≥94%) were from Penta (Prague, Czech Republic). Iron (III) chloride hexahydrate (97%) was obtained from Merck (Darmstadt, Germany). A deionizing column that contains mixed-bed ion exchange resin, ensuring conductivity below 1 µS/cm, with a standard flow rate and operating pressure, was used to produce deionized water for all the experiments.

### 3.2. Plant Material

Dried CM leaves were purchased from a local market in Karditsa, Greece. Then, the CM leaves were sieved using an Analysette 3 PRO (Fritsch GmbH, Oberstein, Germany), and the average resulting particle size was 355 μm. Powder with a particle size below 400 μm was chosen for the experiments and stored in a freezer at −40 °C until further processing.

### 3.3. Experimental Design

The study employed the RSM with a custom design to optimize extraction conditions for TPC, FRAP, and DPPH antiradical activity. This approach was applied to the PEF extraction process for the dry leaves of CM. Six key independent variables were examined: electric field strength (*E*, kV/cm) as *X*_1_, pulse period (*T*_pulse_, μs) as *X*_2_, pulse duration (*t*_pulse_, μs) as *X*_3_, ethanol concentration in water (*C*, % *v/v*) as *X*_4_, liquid-to-solid ratio (*R*, mL/g) as *X*_5_, and extraction time (*t*, min) as *X*_6_, each tested at three levels—low (−1), medium (0), and high (+1)—as shown in Table 7. To ensure reliability, 30 experimental runs were conducted, including two central points, with each experiment repeated three times and the average values recorded.

To improve the model’s predictive accuracy, stepwise regression was employed to eliminate unnecessary terms, thereby minimizing variance and refining the estimation process. This optimization resulted in a second-order polynomial equation (4) that defines the interactions among the six independent variables:
(4)$$Y_{k}\;=\;\beta_{0}+\sum_{i=1}^{2}\beta_{i}X_{i}+\sum_{i=1}^{2}\beta_{\text{ii}}X_{i}^{2}+\sum_{i=1}^{2}\sum_{j=i+1}^{3}\beta_{\text{ij}}X_{i}X_{j}$$
where the independent variables are denoted by *X_i_* and *X_j_*, and the predicted response variable is defined by *Y_k_*. In the model, the intercept and regression coefficients *β*_0_, *β_i_*, *β_ii_*, and *β_ij_* represent the linear, quadratic, and interaction terms, respectively.

Two stainless steel chambers made by Val-Electronic of Athens, Greece, a mode/arbitrary waveform generator by UPG100 of ELV Elektronik AG of Leer, Germany, a digital oscilloscope by Rigol of Beaverton, Oregon, USA, and a high-voltage power generator by Leybold of LD Didactic GmbH (Hürth, Germany) were used to process the samples utilizing PEF. The stirring extraction (STE) technique was carried out using a stirring hotplate manufactured by Heidolph Instruments GmbH & Co. KG of Schwabach, Germany. The UAE treatment was conducted using an Elmasonic P70H ultrasonication bath, which was manufactured by Elma Schmidbauer GmbH of Singen, Germany. Upon completion of each extraction, samples were subjected to centrifugation for 10 min at 10,000× *g* using a NEYA 16R centrifuge (Remi Elektrotechnik Ltd., Palghar, India). Ultimately, supernatants were gathered and preserved at −40 °C.

### 3.4. Determinations

#### 3.4.1. Total Polyphenol Content (TPC)

The TPC was evaluated through a photometric assay described in a previous study [40]. The results were represented as milligrams of gallic acid equivalents (GAE) per gram of dry weight (dw), calculated using a calibration curve (10–100 mg/L of gallic acid; R^2^ = 0.9996) in water. The samples’ absorbances were measured using a Shimadzu UV-1900i UV/Vis spectrophotometer, which is made in Kyoto, Japan. All analyses were carried out three times, and the average was taken to determine the outcomes.

#### 3.4.2. Ferric-Reducing Antioxidant Power (FRAP) Assay

The procedure for determining the extracts’ antioxidant capacity using the widely-used electron-transfer method is detailed in a prior work [40]. Finding out when the iron oxidation state went from +3 to +2 at 620 nm was the key to this technique. The data were presented as μmol of ascorbic acid equivalents (AAE) per gram of dry weight, and a calibration curve of ascorbic acid (50–500 μM in 0.05 M HCl, R^2^ = 0.9997) was used. All analyses were carried out three times, and the average was taken to determine the outcomes.

#### 3.4.3. DPPH Radical Scavenging Assay

A previously described assay [30] for DPPH scavenging was employed. After mixing 25 μL of adequately diluted sample extract with 975 μL of DPPH solution (100 μmol/L in methanol), the absorbance at 515 nm was measured both immediately and 30 min later. The anti-radical activity of ascorbic acid (100–1000 μmol/L in methanol, R^2^ = 0.9926) was measured using a calibration curve, and the results were presented as μmol of AAE per gram of dry weight. All analyses were carried out three times, and the average was taken to determine the outcomes.

#### 3.4.4. Individual Polyphenols by HPLC-DAD

Based on our earlier research, we identified specific polyphenols from the CM leaves’ extracts using high-performance liquid chromatography coupled with a diode array detector (HPLC-DAD) [30]. Shimadzu Europa GmbH, Duisburg, Germany, supplied the liquid chromatograph (type CBM-20A) and diode array detector (model SPD-M20A) used in this work. Between 200 and 800 nm is the detecting wavelength. Using a Phenomenex Luna C18(2) column (100 Å, 5 μm, 4.6 mm × 250 mm) from Phenomenex Inc. in Torrance, CA, USA, the compounds were injected with a volume of 20 μL and then separated at 40 °C. Both the acetonitrile (B) and water (A) mobile phases contained 0.5 percent formic acid. The gradient program began with a constant value of 10 min, progressed to 40% B after 10 min, 70% B after another 10 min, and finally 40% B. A steady 1 mL/min flow rate for the mobile phase was maintained. The chemicals were identified and then quantified using calibration curves (0–50 μg/mL) by comparing the absorbance spectra and retention times to those of purified standards.

### 3.5. Statistical Analysis

All statistical evaluations were conducted using JMP^®^ Pro 16 software (SAS Institute Inc., Cary, NC, USA) supporting response surface methodology (RSM), regression modeling, and distribution analysis. Normality of data was assessed using the Kolmogorov–Smirnov test. To determine statistically significant differences among treatments, analysis of variance (ANOVA) was performed, followed by Tukey’s HSD multiple comparison test at a significance level of *p* < 0.05. Each extraction protocol was repeated at least twice, and all quantitative analyses were conducted in triplicate. Results are reported as mean ± standard deviation. Additionally, PLS, PCA, MCA, and Pareto plot analysis were applied to evaluate multivariate relationships and identify the most influential extraction parameters.

## 4. Conclusions

This study investigated the extraction of high levels of bioactive compounds from CM leaves using eco-friendly solvents such as ethanol and water, in conjunction with the environmentally sustainable PEF technique. A variety of significant extraction parameters that influence yield were examined to identify those that result in an extract abundant in bioactive compounds. The optimal conditions were found to be 19% *v/v* aqueous ethanol, a liquid-to-solid ratio of 70 mL/g, and the following PEF-related parameters: 1 kV/cm electric field strength, a pulse period of 1000 μs, a pulse duration of 75 μs, and a treatment time of 10 min. Compared to other techniques, such as UAE, the application of PEF appears to be more efficient. PEF may also be applied as a pretreatment to conventional techniques like STE, offering a pathway toward enhanced extraction yields and compound recovery. The results of this experimental study emphasize the significance of optimized extraction methods and reveal numerous beneficial compounds in CM leaves. The significant increase in bioactive content allows for the production of powerful CM leaf extracts that could be utilized in cosmetic formulations, functional meals, and medicinal remedies. Further investigation into the scalability and commercial integration of PEF extraction will support the valorization of CM leaves in functional and pharmaceutical products.

## Figures and Tables

**Figure 1 plants-14-02262-f001:**
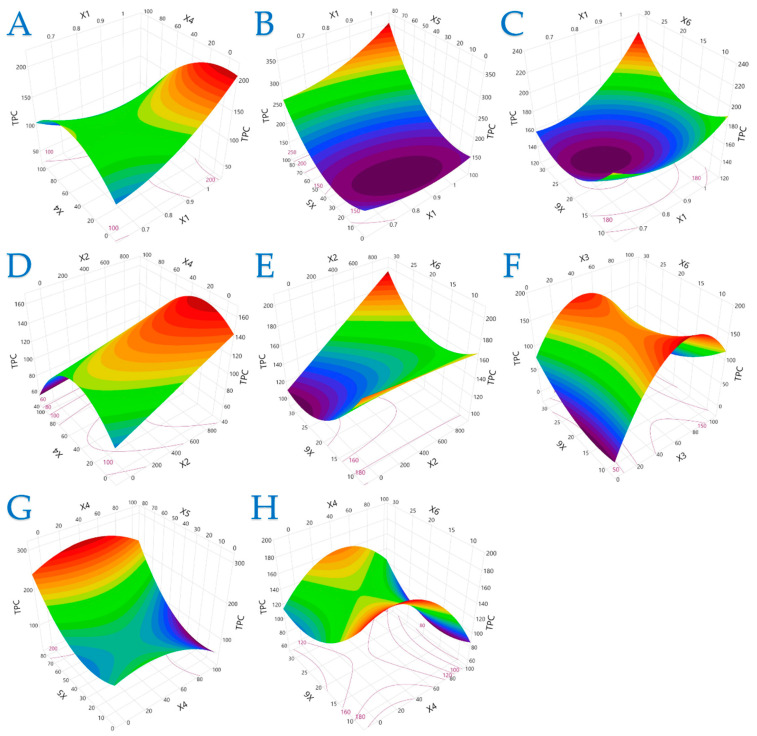
For TPC (mg GAE/g dw), plot (**A**) represents the interaction of *X*_1_ (electric field strength) and *X*_4_ (solvent concentration); plot (**B**) shows the interaction of *X*_1_ and *X*_5_ (liquid-to-solid ratio); plot (**C**) illustrates the interaction of *X*_1_ and *X*_6_ (extraction time); plot (**D**) shows the interaction of *X*_2_ (pulse period) and *X*_4_; plot (**E**) presents the interaction of *X*_2_ and *X*_6_; plot (**F**) illustrates the interaction of *X*_3_ (pulse duration) and *X*_6_; plot (**G**) represents the interaction of *X*_4_ and *X*_5_; and plot (**H**) depicts the interaction of *X*_4_ and *X*_6_.

**Figure 2 plants-14-02262-f002:**
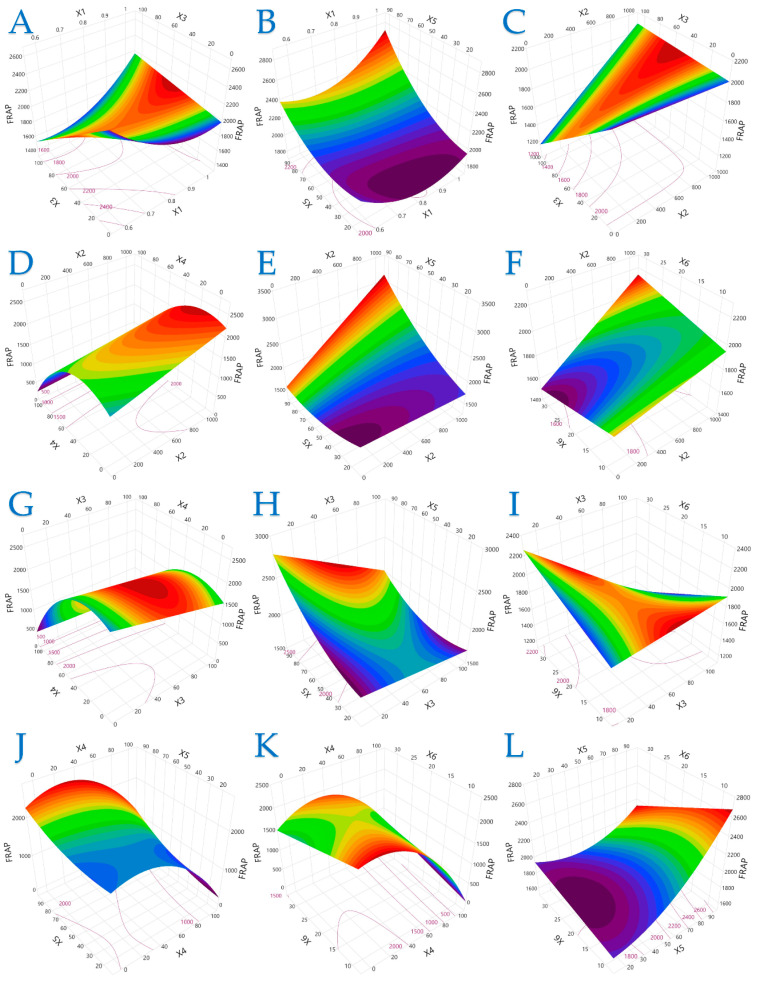
For FRAP (μmol AAE/g dw), plot (**A**) represents the interaction of *X*_1_ (electric field strength) and *X*_3_ (pulse duration); plot (**B**) shows the interaction of *X*_1_ and *X*_5_ (liquid-to-solid ratio); plot (**C**) illustrates the interaction of *X*_2_ (pulse period) and *X*_3_; plot (**D**) shows the interaction of *X*_2_ and *X*_4_ (solvent concentration); plot (**E**) presents the interaction of *X*_2_ and *X*_5_; plot (**F**) illustrates the interaction of *X*_2_ and *X*_6_ (extraction time); plot (**G**) represents the interaction of *X*_3_ and *X*_4_; plot (**H**) depicts the interaction of *X*_3_ and *X*_5_; plot (**I**) illustrates the interaction of *X*_3_ and *X*_6_; plot (**J**) represents the interaction of *X*_4_ and *X*_5_; plot (**K**) illustrates the interaction of *X*_4_ and *X*_6_; and plot (**L**) illustrates the interaction of *X*_5_ and *X*_6_.

**Figure 3 plants-14-02262-f003:**
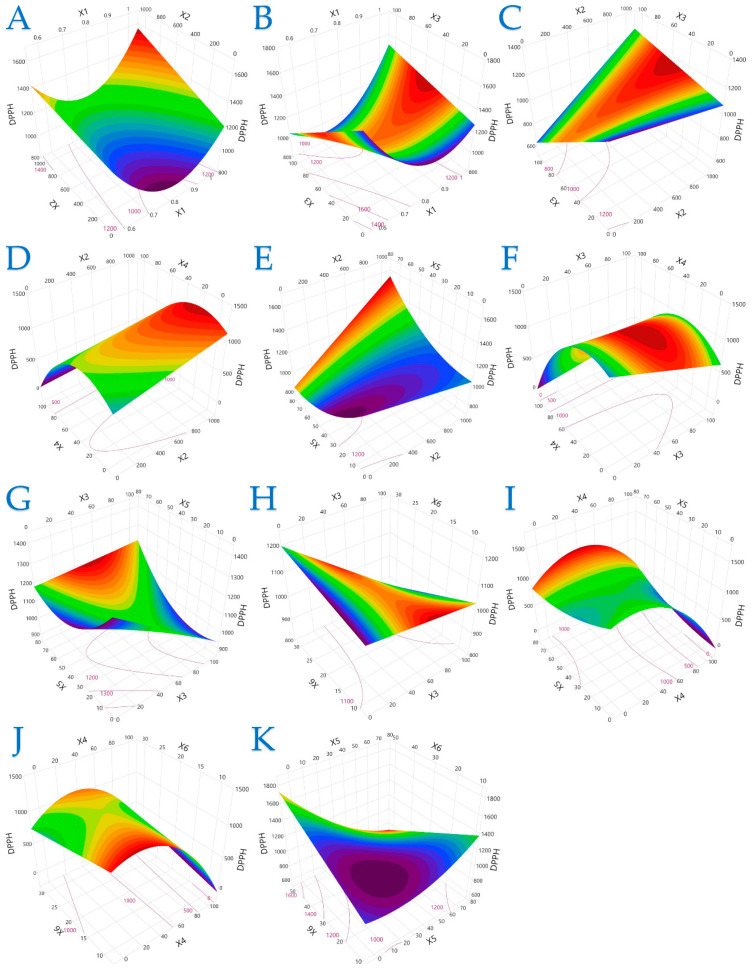
For DPPH (μmol AAE/g dw), plot (**A**) represents the interaction of *X*_1_ (electric field strength) and *X*_2_ (pulse period); plot (**B**) shows the interaction of *X*_1_ and *X*_3_ (pulse duration); plot (**C**) illustrates the interaction of *X*_2_ and *X*_3_; plot (**D**) shows the interaction of *X*_2_ and *X*_4_ (solvent concentration); plot (**E**) presents the interaction of *X*_2_ and *X*_5_ (liquid-to-solid ratio); plot (**F**) illustrates the interaction of *X*_3_ and *X*_4_; plot (**G**) represents the interaction of *X*_3_ and *X*_5_; plot (**H**) depicts the interaction of *X*_3_ and *X*_6_ (extraction time); plot (**I**) illustrates the interaction of *X*_4_ and *X*_5_; plot (**J**) represents the interaction of *X*_4_ and *X*_6_; and plot (**K**) illustrates the interaction of *X*_5_ and *X*_6_.

**Figure 4 plants-14-02262-f004:**
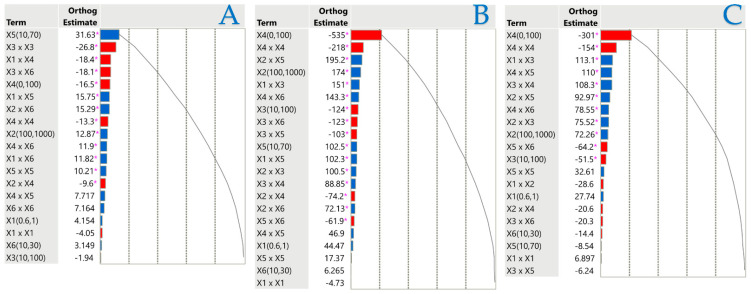
Pareto graphs reveal the importance of each parameter estimate in the stirring extraction method for TPC (**A**), FRAP (**B**), and DPPH tests (**C**). The graphs denote statistical significance with a pink asterisk (*p* < 0.05), with positive values illustrated by blue bars and negative values by red bars.

**Figure 5 plants-14-02262-f005:**
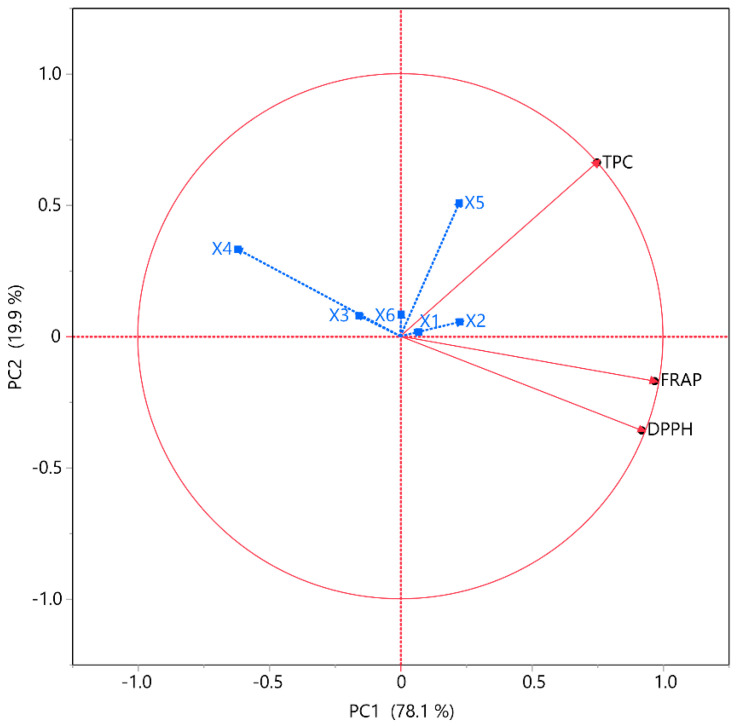
PCA for the assessed variables. Each *X* variable is depicted in blue.

**Figure 6 plants-14-02262-f006:**
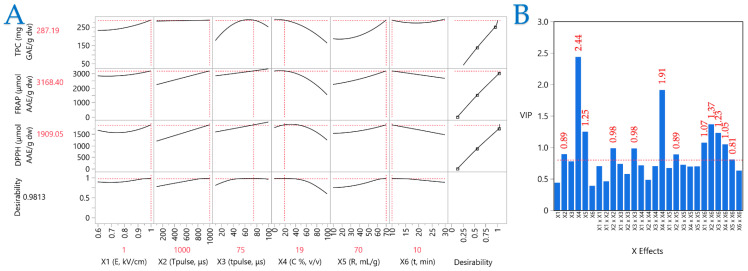
Plot (**A**) shows a Partial Least Squares (PLS) prediction profiler and a desirability function and plot (**B**) illustrates the Variable Importance Plot (VIP).

**Figure 7 plants-14-02262-f007:**
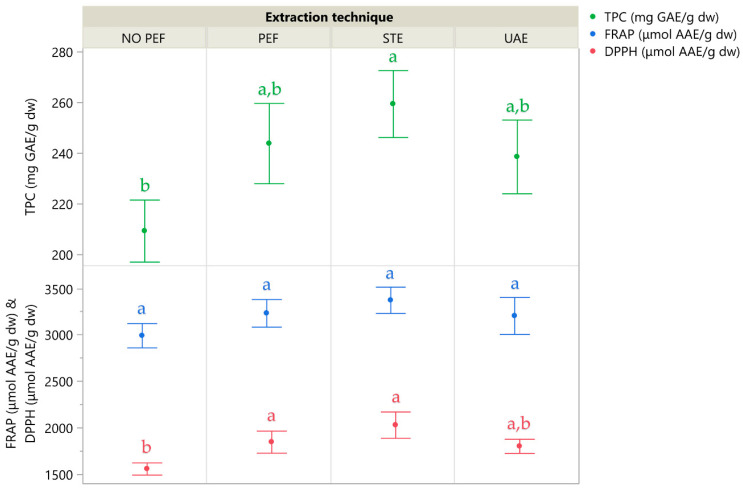
A comparative graph illustrating various extraction techniques, where the mean values are presented alongside error bars that represent standard deviations derived from three independent replicates (*n* = 3). Additionally, lowercase letters (e.g., a, b) indicate statistically significant differences between means (*p* < 0.05).

**Figure 8 plants-14-02262-f008:**
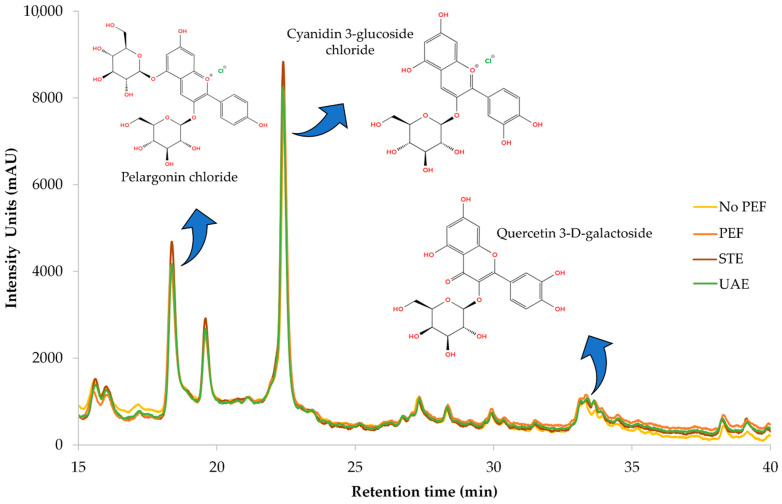
Representative HPLC chromatograms at 280 nm of the optimized dry CM leaves obtained through different extraction techniques, illustrating the identified polyphenolic compounds.

**Table 1 plants-14-02262-t001:** The experimental results elucidate the links between the six independent factors analyzed and the associated responses of the dependent variables.

Design Point	Independent Variables						Actual Responses *		
	*X*_1_ (*E*, kV/cm)	*X*_2_ (*T*_pulse_, μs)	*X*_3_ (*t*_pulse_, μs)	*X*_4_ (*C* %, *v*/*v*)	*X*_5_ (*R*, mL/g)	*X*_6_ (*t*, min)	TPC	FRAP	DPPH
1	1 (1)	0 (550)	1 (100)	−1 (0)	0 (40)	−1 (10)	174.62 ± 6.29	2448.8 ± 134.68	1317.42 ± 44.79
2	−1 (0.6)	1 (1000)	−1 (10)	0 (50)	1 (70)	1 (30)	176.87 ± 12.91	2889.34 ± 86.68	1421.42 ± 48.33
3	0 (0.8)	1 (1000)	1 (100)	−1 (0)	1 (70)	0 (20)	148.87 ± 7	2412.86 ± 132.71	1197.43 ± 50.29
4	1 (1)	1 (1000)	−1 (10)	1 (100)	1 (70)	−1 (10)	42.3 ± 2.2	922.05 ± 23.97	537.19 ± 13.97
5	−1 (0.6)	−1 (100)	1 (100)	−1 (0)	−1 (10)	−1 (10)	135.94 ± 5.71	2049.16 ± 45.08	1054.4 ± 61.16
6	−1 (0.6)	1 (1000)	−1 (10)	1 (100)	−1 (10)	0 (20)	47.75 ± 1.24	655.11 ± 46.51	333.73 ± 16.35
7	0 (0.8)	0 (550)	0 (55)	0 (50)	0 (40)	0 (20)	117.46 ± 8.46	1795.53 ± 79	968.18 ± 49.38
8	0 (0.8)	0 (550)	0 (55)	0 (50)	0 (40)	0 (20)	110.08 ± 2.31	1657.12 ± 67.94	886.99 ± 62.09
9	−1 (0.6)	−1 (100)	1 (100)	−1 (0)	1 (70)	1 (30)	22.47 ± 0.45	310.38 ± 16.14	144.8 ± 8.11
10	1 (1)	0 (550)	1 (100)	0 (50)	1 (70)	1 (30)	191.36 ± 11.48	1963.65 ± 100.15	1356.99 ± 74.63
11	1 (1)	0 (550)	0 (55)	1 (100)	−1 (10)	−1 (10)	40.8 ± 2.77	348.12 ± 20.19	189.42 ± 11.37
12	−1 (0.6)	1 (1000)	0 (55)	0 (50)	0 (40)	−1 (10)	194.62 ± 11.29	2112.07 ± 101.38	1378.5 ± 100.63
13	0 (0.8)	1 (1000)	1 (100)	1 (100)	0 (40)	−1 (10)	32.97 ± 1.32	758.98 ± 44.78	470 ± 17.86
14	1 (1)	−1 (100)	−1 (10)	1 (100)	1 (70)	1 (30)	130.13 ± 5.99	1180.18 ± 44.85	419.76 ± 23.51
15	1 (1)	1 (1000)	−1 (10)	−1 (0)	1 (70)	1 (30)	245.4 ± 7.85	2716.59 ± 116.81	1233.19 ± 91.26
16	0 (0.8)	1 (1000)	0 (55)	1 (100)	0 (40)	1 (30)	147.29 ± 5.3	1033.19 ± 34.1	549.16 ± 36.24
17	−1 (0.6)	0 (550)	0.27 (67.15)	1 (100)	1 (70)	0 (20)	184.65 ± 13.11	971.9 ± 60.26	811.52 ± 25.16
18	0 (0.8)	−1 (100)	0 (55)	0 (50)	1 (70)	−1 (10)	250.63 ± 14.29	1802.34 ± 133.37	1091.95 ± 34.94
19	−1 (0.6)	−1 (100)	1 (100)	1 (100)	−1 (10)	1 (30)	24.78 ± 1.54	624.49 ± 19.98	411.54 ± 30.04
20	1 (1)	−1 (100)	1 (100)	1 (100)	0 (40)	0 (20)	23.03 ± 1.06	661.54 ± 21.17	524.84 ± 28.87
21	1 (1)	−1 (100)	1 (100)	−1 (0)	−1 (10)	1 (30)	60.31 ± 2.05	1640.16 ± 60.69	1097.68 ± 24.15
22	−1 (0.6)	1 (1000)	1 (100)	−1 (0)	−1 (10)	1 (30)	70.92 ± 4.04	1804.52 ± 77.59	1024.02 ± 35.84
23	0 (0.8)	0 (550)	−1 (10)	0 (50)	0 (40)	0 (20)	87.62 ± 4.12	2223.42 ± 106.72	1283.76 ± 38.51
24	−1 (0.6)	−1 (100)	−1 (10)	1 (100)	0 (40)	−1 (10)	24.56 ± 0.54	819.87 ± 44.27	551.28 ± 34.73
25	1 (1)	1 (1000)	1 (100)	0 (50)	−1 (10)	0 (20)	85.36 ± 2.05	2040.18 ± 112.21	1454.52 ± 72.73
26	1 (1)	0 (550)	−1 (10)	0 (50)	−1 (10)	1 (30)	85.87 ± 2.66	2036.2 ± 148.64	1416.55 ± 89.24
27	−1 (0.6)	0 (550)	−1 (10)	−1 (0)	1 (70)	−1 (10)	92.4 ± 3.42	2814.98 ± 137.93	1607 ± 61.07
28	0 (0.8)	1 (1000)	−1 (10)	−1 (0)	−1 (10)	−1 (10)	72.67 ± 3.49	1888.21 ± 84.97	1383.27 ± 45.65
29	−1 (0.6)	−1 (100)	0 (55)	−1 (0)	−1 (10)	1 (30)	80.8 ± 4.2	1959.68 ± 84.27	1501.49 ± 58.56
30	1 (1)	−1 (100)	−1 (10)	−1 (0)	0 (40)	−1 (10)	83.3 ± 4.75	2196.45 ± 103.23	1404.91 ± 105.37

* Values represent the mean of triplicate determinations; TPC, total polyphenol content (mg GAE/g dw); FRAP, ferric-reducing antioxidant power (μmol AAE/g dw); DPPH, antiradical activity (μmol AAE/g dw).

**Table 2 plants-14-02262-t002:** The variance analysis (ANOVA) for the quadratic polynomial model used in the response surface methodology.

Factor	TPC	FRAP	DPPH
Stepwise regression			
Intercept	138.2 *	1832 *	1043 *
*X*_1_—electric field strength	5.769	13.29	6.376
*X*_2_—pulse period	14.73 *	173.3 *	96.03 *
*X*_3_—pulse duration	12.73 *	−172 *	−82.6 *
*X*_4_—solvent concentration	−23.6 *	−642 *	−357 *
*X*_5_—liquid-to-solid ratio	39.99 *	112.7 *	14.11
*X*_6_—extraction time	−3.5	−8.24	−12.9
*X* _1_ ^2^	14.86	139.4 *	212.1 *
*X* _1_ *X_2_*	-	-	40.99
*X* _2_ ^2^	-	-	-
*X* _1_ *X* _3_	-	199.8 *	145.9 *
*X* _2_ *X* _3_	-	168 *	132.2 *
*X* _3_ ^2^	−72.5 *	-	-
*X* _1_ *X* _4_	−27.4 *	-	-
*X* _2_ *X* _4_	−7.07	−83.8 *	−33.1
*X* _3_ *X* _4_	-	87.38 *	152.7 *
*X* _4_ ^2^	−32 *	−518 *	−370 *
*X* _1_ *X* _5_	16.39 *	73.04	-
*X* _2_ *X* _5_	-	253.7 *	142.9 *
*X* _3_ *X* _5_	-	−77	72.55 *
*X* _4_ *X* _5_	10.35	85.32 *	174.7 *
*X* _5_ ^2^	32.25 *	89.7	74.88
*X* _1_ *X* _6_	11.36	-	-
*X* _2_ *X* _6_	16.4 *	72.42 *	-
*X* _3_ *X* _6_	−24 *	−174 *	−45.1
*X* _4_ *X* _6_	17.16 *	217.2 *	125.2 *
*X* _5_ *X* _6_	-	−102 *	−106 *
*X* _6_ ^2^	18.65	-	-
*ANOVA*			
*F*-value (model)	15.77	58.95	26.33
*F*-value (lack of fit)	16.71	1.46	3.52
*p*-value (model)	<0.0001 *	<0.0001 *	<0.0001 *
*p*-value (lack of fit)	0.1878 ^ns^	0.5692 ^ns^	0.3919 ^ns^
*R* ^2^	0.968	0.994	0.983
Adjusted *R*^2^	0.906	0.977	0.946
RMSE	20.31	114.9	103.3
MR	106.2	1625	957.4
CV	62.47	46.34	45.88
DF (total)	29	29	29

* Values significantly affected responses at a probability level of 95% (*p* < 0.05). TPC, total polyphenol content; FRAP, ferric-reducing antioxidant power; DPPH, antiradical activity; ^ns^, non-significant; *F*-value, test for comparing model variance with residual (error) variance; *p*-value, probability of seeing the observed *F*-value if the null hypothesis is true; RMSE, root mean square error; MR, mean of response; CV, coefficient of variation; DF, degrees of freedom.

**Table 3 plants-14-02262-t003:** Optimal extraction conditions and maximum anticipated responses for the dependent variables.

Parameters	*X*_1_ (*E*, kV/cm)	*X*_2_ (*T*_pulse_, μs)	*X*_3_ (*t*_pulse_, μs)	*X*_4_ (*C* %, *v*/*v*)	*X*_5_ (*R*, mL/g)	*X*_6_ (*t*, min)	Desirability	Stepwise Regression
TPC (mg GAE/g dw)	1	1000	51	26	70	30	0.9978	312.86 ± 40.19
FRAP (μmol AAE/g dw)	0.6	1000	10	5	70	10	0.9930	3089.42 ± 259.61
DPPH (μmol AAE/g dw)	0.6	310	10	16	10	22	0.9980	2021.39 ± 225.06

**Table 4 plants-14-02262-t004:** Multivariate correlation analysis of assessed variables.

Responses	TPC	FRAP	DPPH
TPC	‒	0.6043	0.4574
FRAP		‒	0.9221
DPPH			‒

**Table 5 plants-14-02262-t005:** Optimal extraction conditions for polyphenolic compounds obtained through different extraction techniques, applied to the dry plant CM extracts.

Compound	No-PEF	PEF	STE	UAE
Pelargonin chloride	110.66 ± 1.71 ^c^	162.44 ± 6.66 ^a^	174.47 ± 4.71 ^a^	142.32 ± 8.25 ^b^
Cyanidin 3-glucoside chloride	6.21 ± 0.04 ^c^	8.86 ± 0.33 ^a^	9.16 ± 0.2 ^a^	8.18 ± 0.2 ^b^
Quercetin 3-*D*-galactoside	3.02 ± 0.09 ^a^	3.26 ± 0.14 ^a^	3.18 ± 0.1 ^a^	3.19 ± 0.24 ^a^
Total identified	119.89 ± 1.84 ^c^	174.55 ± 7.13 ^a^	186.81 ± 5.01 ^a^	153.7 ± 8.69 ^b^

Each method’s values reflect the average ± the standard deviation (SD) of three separate experiments. The presence of different superscript letters (for example, a–c) within each line signifies statistically significant differences (*p* < 0.05).

**Table 6 plants-14-02262-t006:** Calibration curve equations for each compound detected via HPLC-DAD.

Polyphenolic Compound	Equation (Linear)	R^2^	RT (min)	UV_max_	LOD (mg/L)	LOQ (mg/L)
Pelargonin chloride	y = 1610.01x − 2626.92	0.997	18.900	275	2.84	8.61
Cyanidin 3-glucoside chloride	y = 46,680.57x − 10.63	0.999	22.312	516	0.97	2.94
Quercetin 3-*D*-galactoside	y = 41,489.69x − 35,577.55	0.993	33.598	257	3.96	12.00

**Table 7 plants-14-02262-t007:** Optimal process parameters, including both the actual and coded values of the independent variables.

Independent Variables	Code Units	Coded Variable Level		
		−1	0	1
Electric field strength (*E*, kV/cm)	*X* _1_	0.6	0.8	1
Pulse period (*T*_pulse_, μs)	*X* _2_	100	550	1000
Pulse duration (*t*_pulse_, μs)	*X* _3_	10	55	100
Solvent concentration (*C* %, *v*/*v*)	*X* _4_	0	50	100
Liquid-to-solid ratio (*R*, mL/g)	*X* _5_	10	40	70
Extraction time (*t*, min)	*X* _6_	10	20	30

## Data Availability

The original contributions presented in this study are included in the article. Further inquiries can be directed to the corresponding author.

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
