# Peer review of "Isolation of Bioactive Compounds and Antioxidant Activity Evaluation of Crataegus monogyna Leaves via Pulsed Electric Field-Assisted Extraction"

_plants, 2025, doi:10.3390/plants14152262_

Round 1
Reviewer 1 Report
Comments and Suggestions for Authors
This study explores the application of pulsed electric field (PEF) for enhanced extraction of bioactive compounds from C. monogyna leaves by investigating liquid-to-solid ratio, solvent composition, and pulse duration, pulse period, electric field intensity, and treatment duration during the optimization process. The manuscript contains very big data of figures and tables, while with rough description and discussion of the results. Major revisions are recommended, questions and suggestions are as follows.
1 for the levels of the independent variables used to optimize the process in Table 7, why three levels as 100, 550, 1000 μs, were designed for pulse period, and 10,55, 100μs, were designed for pulse duration?
2 line 338-341, for the control as the conventional and modern extraction, what are the detailed extraction conditions applied, and did you refer to any references?
3 for the vast antioxidant assays, why were TPC, FRAP, DPPH assays applied as actual responses for the optimization of extraction? And did you carry out repeated experiments, with no standard deviations in the results in Table 1?
4 for figure 1-3, more than ten figures in each of them, however, so few descriptions could be seen without any presentations like figure 1(A), figure1 (B)…….. the same problems are also need to reconsidered in other figures and tables.
5 figure captions should be reorganized, for example figure 6, the caption description like this could be just provided in the text.
6 for the importance of Statistical Analysis in this research, the detailed methods and potential roles for the performed PLS, Pareto plot analysis, PCA, and MCA should be specified.
Author Response
This study explores the application of pulsed electric field (PEF) for enhanced extraction of bioactive compounds from C. monogyna leaves by investigating liquid-to-solid ratio, solvent composition, and pulse duration, pulse period, electric field intensity, and treatment duration during the optimization process. The manuscript contains very big data of figures and tables, while with rough description and discussion of the results. Major revisions are recommended, questions and suggestions are as follows.
We would like to thank the reviewer for his/her valuable comments.
1 for the levels of the independent variables used to optimize the process in Table 7, why three levels as 100, 550, 1000 μs, were designed for pulse period, and 10,55, 100μs, were designed for pulse duration?
The levels of pulse period (100, 550, 1000 μs) and pulse duration (10, 55, 100 μs) were selected based on relevant literature and supported by preliminary trials, which indicated that these values yield meaningful variations in extraction efficiency.
2 line 338-341, for the control as the conventional and modern extraction, what are the detailed extraction conditions applied, and did you refer to any references?
We appreciate the reviewer’s interest in the clarity of our control and comparative methods. The manuscript has been updated to include a dedicated subsection titled “Comparative Analysis of Extraction Techniques”, where the procedures and rationale for all comparative extractions are described in detail.
3 for the vast antioxidant assays, why were TPC, FRAP, DPPH assays applied as actual responses for the optimization of extraction? And did you carry out repeated experiments, with no standard deviations in the results in Table 1?
The aim of this work is to optimize the recovery of bioactive compounds with antioxidant activity from hawthorn leaves. Therefore, it is logical to use TPC, FRAP and DPPH as the main tethers for the optimization. TPC, FRAP, and DPPH are well-established assays commonly used to evaluate antioxidant activity, which is central to the scope of our optimization. All experiments and analyses were conducted in triplicate. Although RSM typically presents mean values, we have now added standard deviations to Table 1 as requested for transparency.
4 for figure 1-3, more than ten figures in each of them, however, so few descriptions could be seen without any presentations like figure 1(A), figure1 (B)…….. the same problems are also need to reconsidered in other figures and tables.
Additional explanation has been added to support each Figure 1 to 3 and Table 3.
5 figure captions should be reorganized, for example figure 6, the caption description like this could be just provided in the text.
Figure captions have been revised for clarity. Descriptive content previously included in captions, such as that of Figure 6, has now been moved to the main text where appropriate.
6 for the importance of Statistical Analysis in this research, the detailed methods and potential roles for the performed PLS, Pareto plot analysis, PCA, and MCA should be specified.
Detailed explanations for all applied statistical methods—including PLS, Pareto plots, PCA, and MCA—have been added to the methodology and results sections to clarify their role in optimizing the extraction process and interpreting results.
Reviewer 2 Report
Comments and Suggestions for Authors
Line 37: „such as fruits (berries, watermelon, apples, grapes), vegetables (soybeans, onions)” – please add etc. after the examples or e.g. before: „such as fruits (e.g. berries, watermelon, apples, grapes), vegetables (e.g. soybeans, onions)”
Line 56-63: Oligomeric procyanidins are one of the most important active compounds of C. monogyna, why are they missing form the list of constituents?
Line 59: “alkaloids (caffeic acid, amygdalin),” – Are you sure that C. monogyna contains any alkaloids? The given references do not mention it. Caffeic acid is a cinnamic acid derivative, a type of phenolic acids, and a polyphenol, not an alkaloid. And amygdalin is classified as a cyanogenic glycoside.
Line 255-259: I do not have access to the methodology of the mentioned source (Bahorun et al. [28]), please check whether they also applied the same reference compounds (GAE, AAE) in their measurement results to, and indicate in the article.
Line 283: “anthocyanin found in berry fruits like mulberries and strawberries [31,32].” Ref [32] deals with mulberries, and I couldn’t find strawberries in ref [31] either. Are you sure that the sentence and the references are correct?
Author Response
Line 37: „such as fruits (berries, watermelon, apples, grapes), vegetables (soybeans, onions)” – please add etc. after the examples or e.g. before: „such as fruits (e.g. berries, watermelon, apples, grapes), vegetables (e.g. soybeans, onions)”
We would like to thank the reviewer for his/her valuable comments. We have modified the text to include ‘e.g.’ before the examples, as suggested, to reflect that these are representative items.
Line 56-63: Oligomeric procyanidins are one of the most important active compounds of C. monogyna, why are they missing form the list of constituents?
We acknowledge the omission of oligomeric procyanidins. Their absence may be attributed to unsuccessful isolation or degradation during the extraction process. This limitation has been noted in the revised discussion.
Line 59: “alkaloids (caffeic acid, amygdalin),” – Are you sure that C. monogyna contains any alkaloids? The given references do not mention it. Caffeic acid is a cinnamic acid derivative, a type of phenolic acids, and a polyphenol, not an alkaloid. And amygdalin is classified as a cyanogenic glycoside.
We thank the reviewer for pointing out this important clarification. The term “alkaloids” was incorrectly applied to compounds such as caffeic acid and amygdalin. In response, the entire phrase was removed from the manuscript to avoid misrepresentation. We have carefully reviewed the classification of all listed compounds to ensure accuracy in the revised version.
Line 255-259: I do not have access to the methodology of the mentioned source (Bahorun et al. [28]), please check whether they also applied the same reference compounds (GAE, AAE) in their measurement results to, and indicate in the article.
We have verified that Bahorun et al. used the same reference compounds (GAE and AAE) in their measurements. This has been clarified in the revised manuscript.
Line 283: “anthocyanin found in berry fruits like mulberries and strawberries [31,32].” Ref [32] deals with mulberries, and I couldn’t find strawberries in ref [31] either. Are you sure that the sentence and the references are correct?
We corrected the reference to ensure consistency with the statement. The cited studies have been cross-checked to accurately reflect the presence of anthocyanins in the mentioned berries.
Reviewer 3 Report
Comments and Suggestions for Authors
The choice of the anthocyanins pelargonidin and cyanidin-3-glucoside as marker compounds appears unusual. Nowhere in the introduction is it mentioned that anthocyanins are typical metabolites found in hawthorn leaves, which makes it necessary to provide relevant references. Please include the UV-VIS spectra of the analytes determined by HPLC-DAD. In the presented chromatogram (Fig. 8), the signal is recorded at 280 nm — a wavelength that is not specific for either anthocyanins or hyperoside. At which wavelengths were pelargonidin, cyanidin-3-glucoside, and hyperoside detected? Was the method validated? If so, please include the validation results or cite a source that provides detailed information on the validation procedure. Additionally, why was the content of vitexin not determined?
Author Response
The choice of the anthocyanins pelargonidin and cyanidin-3-glucoside as marker compounds appears unusual. Nowhere in the introduction is it mentioned that anthocyanins are typical metabolites found in hawthorn leaves, which makes it necessary to provide relevant references. Please include the UV-VIS spectra of the analytes determined by HPLC-DAD. In the presented chromatogram (Fig. 8), the signal is recorded at 280 nm — a wavelength that is not specific for either anthocyanins or hyperoside. At which wavelengths were pelargonidin, cyanidin-3-glucoside, and hyperoside detected?8 Was the method validated? If so, please include the validation results or cite a source that provides detailed information on the validation procedure. Additionally, why was the content of vitexin not determined?
We would like to thank the reviewer for his/her valuable comments.
Anthocyanins were not initially highlighted in the introduction as characteristic compounds of hawthorn leaves. We have now included references supporting the presence and relevance of pelargonidin and cyanidin-3-glucoside, as marker compounds for antioxidant studies.
We have added UV-Vis spectra (see Figure A1) for pelargonidin, cyanidin-3-glucoside, and hyperoside, as detected by HPLC-DAD. Each compound was quantified at its maximum absorbance wavelength (UVmax); although peaks were visible at multiple wavelengths, 280 nm was selected for presentation due to visual clarity.
The analytical method was validated, and the results have now been included in the revised methodology section. Vitexin and its derivatives were not quantified due to the absence of appropriate standards.
Round 2
Reviewer 1 Report
Comments and Suggestions for Authors
The authors have addressed all my comments, and the manuscript has been improved greatly for acceptance.
Author Response
The authors have addressed all my comments, and the manuscript has been improved greatly for acceptance.
Thank you for your positive feedback. We appreciate your careful review and are pleased that the revised manuscript meets your expectations.
Reviewer 3 Report
Comments and Suggestions for Authors
The changes are not marked in the submitted manuscript. Please provide a version with all modifications highlighted.
Author Response
The changes are not marked in the submitted manuscript. Please provide a version with all modifications highlighted.
Thank you for your feedback. We would like to clarify that the revised manuscript had indeed been uploaded with all modifications visible through track changes. However, a clean version without changes was uploaded from the journal office. To ensure full transparency, we have now re-uploaded the revised paper with all modifications clearly highlighted using yellow shading for your convenience. We apologize for any confusion this may have caused.